
# The water vapor self-continuum absorption in the infrared atmospheric windows:
# New laser measurements near 3.3 µm and 2.0 µm

Loic Lechevallier[1,2], Semen Vasilchenko[1,3], Roberto Grilli[2], Didier Mondelain[1], Daniele Romanini[1] and Alain Campargue[1]

[1]Univ. Grenoble Alpes, CNRS, LIPhy, 38000 Grenoble, France
[2] Univ. Grenoble Alpes, CNRS, IRD, Grenoble INP*, IGE, F-38000 Grenoble, France[1]
[3] Laboratory of Molecular Spectroscopy, V.E. Zuev Institute of Atmospheric Optics, SB, Russian Academy of Science, 1 AkademicianZuev square, 634021 Tomsk, Russia

*Correspondence to*: Alain Campargue (Alain.Campargue@univ-grenoble-alpes.fr)

**Abstract.** The amplitude, the temperature dependence and the physical origin of the water vapor absorption continuum are a long standing issue in molecular spectroscopy with direct impact in atmospheric and planetary sciences. In the recent years, we have determined the self-continuum absorption of water vapor at different spectral points of the atmospheric windows at 4.0, 2.1, 1.6 and 1.25 µm, by highly sensitive cavity enhanced laser techniques. These accurate experimental constraints have been used to adjust the last version (V3.2) of the semi-empirical MT_CKD model (Mlawer-Tobin_Clough-Kneizys-Davies) widely incorporated in atmospheric radiative transfer codes. In the present work, the self-continuum cross-sections, $C_S$, are newly determined at 3.3 µm (3007 cm$^{-1}$) and 2.0 µm (5000 cm$^{-1}$) by optical-feedback-cavity enhanced absorption spectroscopy (OFCEAS) and cavity ring-down spectroscopy (CRDS), respectively. These new data allow completing the spectral coverage of the 4.0 and 2.1 µm windows, respectively, and testing the recently released V3.2 version of the MT_CKD3 continuum. By complementing high temperature literature data to the present data, the temperature dependence of the self continuum is presented.

---

[1]Institute of Engineering Univ. Grenoble Alpes



## 1. Introduction

About 60% of the solar radiation in traversing the Earth's atmosphere is absorbed by water vapor. Water vapor absorption involves two contributions: *(i)* a succession of vibrational bands located every 1500-2000 cm$^{-1}$ with intensity decreasing from the far infrared to the visible. These are formed by multitude of narrow rovibrational absorption lines which are

tabulated in spectroscopic databases like HITRAN (Gordon et al., 2017) and GEISA (Jacquinet-Husson et al., 2016), *(ii)* a weak broadband absorption continuum with slow frequency dependence roughly following the vibrational bands. In the regions of low absorption between the vibrational bands - the water vapor transparency windows- the continuum has a major contribution and may dominate the cumulative effect of the weak rovibrational lines. In the Earth's atmosphere, the continuum decomposes into a self-continuum component due to water-water molecular interactions and a foreign-continuum

component due to water-nitrogen and water-oxygen interactions. The water self-continuum increases quadratically with the water number density and has strong negative temperature dependence. More than one century after its discovery (Rubens and E. Aschkinass, 1898) and decades of controversy (Shine et al., 2012), the origin of the water continuum remains an open question. The reader is referred to (Camy-Peyret et al., 2003; Ma et al., 2008; Ptashnik et al., 2008, 2011b; Leforestier et al., 2010, Tretyakov et al., 2014; Buryak and Vigasin, 2015; Vigasin, 2014; Serov et al. 2017) for discussions about the

respective contributions of the far wings of the rovibrational lines, the water dimers (stable and metastable) and collision-induced absorption.

The water vapor continuum implemented in atmospheric radiative transfer codes is usually the semi-empirical MT_CKD model (Mlawer-Tobin_Clough-Kneizys-Davies) (Clough et al. 1989; Mlawer et al. 2012) which is regularly updated. This model which is basically a far wing line shape model in the window regions involves a number of *ad hoc* parameters which

have been constrained to laboratory and field measurements mostly in the long- and mid-infrared. Because of the experimental difficulty in detecting very small variation of the spectrum baseline induced by the injection of water vapor in an absorption cell (for instance 10$^{-8}$ cm$^{-1}$), up to recently, measurements were very scarce in the mid- and near-infrared and the MT_CKD model used long range extrapolations with a few experimental constraints dating back to the 1980s (e.g. Burch et al., 1982, 1984, 1985). Thus, a validation of the amplitude and of the temperature-dependence of the MT_CKD continuum

in those spectral ranges is highly desirable not only to quantify its impact on the estimated radiative budget of the Earth (Held and Soden, 2000; Ptashnik et al., 2011a; Paynter and Ramaswamy, 2011; 2012; 2014) but also on the extension of the habitable zone around stars (Kasting et al., 1993; Kopparapu et al., 2009; Leconte et al. 2013a; 2013b).

In the last decade, large experimental efforts have been undertaken in different laboratories to improve the knowledge of the water continuum in the transparency windows. As a result, coinciding measurements obtained by Fourier Transform

Spectroscopy (FTS) with long path absorption cells (Paynter et al., 2009; Ptashnik et al., 2011a; 2013; 2015) were found in strong disagreement with the MT_CKDV2.5 model in the 4.0, 2.1 and 1.6 μm windows. In particular, the reported FTS values are larger by one and two orders of magnitude in the center of the 2.1 and 1.6 μm windows.





More recently, we have implemented highly sensitive cavity enhanced laser techniques at different spectral points of the 4.0, 2.1, 1.6 and 1.25 µm to measure in the laboratory the self-continuum absorption of water vapor (Mondelain et al. 2013; 2014, 2015; Campargue et al. 2016; Ventrillard et al. 2015; Richard et al. 2017). Overall, these measurements by Optical Feedback Cavity Enhanced Absorption Spectroscopy (OFCEAS) and Cavity Ring-down Spectroscopy (CRDS) were found

closer to the MT_CKD model than to the FTS results. Experimentally, such investigations are demanding as, contrary to FTS which allows for large spectral coverage, the tunability of laser sources is generally limited in particular in the mid-infrared ranges, and each spectral point requires dedicated experimental developments. Nevertheless, the sensitivity, base line stability and data acquisition speed provided by CRDS and OFCEAS are critical advantages to avoid experimental biases and retrieve accurate continua. In all our previous measurements and contrary to the above mentioned FTS studies, the

pressure squared dependence of the continuum was carefully checked during pressure ramps to insure the gas phase origin of the measured signals and exclude experimental biases related to the stability of the spectrometer or to the impact of water on the mirror reflectivity.

In the present work, we report two new measurement points by OFCEAS at 3.33 µm (3007 cm⁻¹) and CRDS at 2.0 µm (5000 cm⁻¹). These new measurements complete the sampling of the 4.0 µm and 2.1 µm windows at high energy. The temperature

dependence of the OFCEAS study at 3.33 µm is studied in the 298-323 K interval and discussed in relation with literature data at higher temperature. The present new results are compared to the recently released V3.2 version of the MT_CKD model which was adjusted taking into account part of our previous measurements.

The two next sections present for each of the two spectral points, the experimental setups, spectra acquired, the continuum retrieval and a comparison with literature data. Considerations on the temperature dependence are presented in Section 4. An

overview comparison of the experimental results and of the MT_CKD model in the 1500-9000 cm⁻¹ range is presented in the last section

## 2. OFCEAS at 3007 cm⁻¹

OFCEAS is a cavity enhanced absorption method alternative to CRDS (Morville et al., 2004; Romanini et al., 2006; Maisons et al., 2010; Gagliardi and Loock, 2014) which has specific advantages, in particular for trace detection analysis (Kassi et al.

2006; Faïn et al. 2014; Desbois et al. 2014). This technique uses feedback of light from a high-finesse V-cavity to lock the laser emission over cavity resonances during a laser frequency scan. Every second, several spectra over a small spectral interval (~ 1 cm⁻¹) are obtained by OFCEAS allowing for an efficient statistical averaging. While CRDS relies on the measurements of the lifetime of the photons in the cavity, OFCEAS provides transmission spectra. Using the photon lifetime measured at the end of each scan, the cavity transmission is converted into a spectrum in absolute absorption coefficient

scale using the procedure detailed in Kerstel et al. (2006) and Richard et al. (2017).





### 2.1 Spectra acquisition

The OFCEAS spectrometer used in this work is similar to that presented in Ventrillard et al., (2015) and Richard et al. (2017). The laser source (from Nanoplus GmbH) is an Interband Cascade Laser (ICL) emitting between 3005.5 and 3009.5 cm$^{-1}$ by acting on the current and temperature. The cavity mirrors (from LohnStar Optics) have a 99.96 % reflectivity as

deduced from a measured ring-down time of 3.20 μs with empty cavity. The acquired spectra have an intrinsically linear frequency scale with equally spaced spectral points separated by the cavity Free Spectral Range, $FSR=c/2nL$, where $c$ is the speed of light, $L$ the effective cavity length, and $n$ the refractive index of the intra-cavity sample. Both arms of the V-cavity have a length of 40 cm leading to an $FSR$ of 187.4±0.5 MHz. A spectrum over 105 cavity modes (or $FSR$) corresponding 0.7 cm$^{-1}$ is acquired within about 100 ms by modulation of the laser current at 10 Hz. The laser scans, acquisition, laser

temperature and cavity temperature and pressure were controlled by an electronic card (from AP2E). The optical assembly is composed by the diode laser, an infrared polarizer (colorPol MIR), two stirring aluminum mirrors, a photodiode (PCI-2TE-4) and the optical cavity (7.75 mm apex diameter). The laser ramp is provided by the software, and the frequency acquisition rate is maximized. The whole assembly is suspended on two anti-vibration multidirectional mounts, and confined in a thermal isolated aluminum case. A Pt1000 thermistance is placed at the center of the monoblock stainless cavity and its

temperature is controlled by two heating bands *via* a PID regulation. In this study, the temperature of the high finesse cell was varied between 25°C and 50°C. Water vapor was generated by using a steel tank filled with few cm$^3$ of deionized water connected to the spectrometer upstream. Series of spectra were recorded in flow regime (typically 0.13 sccm) during pressure ramps up to 23 mbar, in order to remain below the saturation pressure (about 28 mbar at 296 K).

The pressure ramp was adjusted by a manual and a proportional valve placed before and after the cavity,

respectively. About 6000 spectra were recorded during the pressure ramp (about 10 min). The gas pressure was measured upstream the cavity by a CPG2500 sensor (from Mensor, 0-1000 mbar with an accuracy better than 0.5 mbar). A second sensor was directly connected to the cavity. During the data acquisition, we noted significant differences between the pressure values provided by the two sensors, especially for low pressure values indicating the presence of a pressure gradient between the two measurement points. This is due to the tube fitting which causes pressure losses on the line (Venturi effect),

producing pressure artefacts especially when pressure is low or rapidly varying. Therefore, we decided to deduce the actual water vapor pressure (or molecular density) by directly using two strong absorption lines of the main isotope (H$_2^{16}$O) present in the recorded spectra (see below).

The calibration of the frequency axis relies on the known $FSR$ value (187.5 MHz) and on the absolute frequency of water absorption lines as given in the HITRAN2016 database (Gordon et al., 2017). **Fig. 1** shows typical spectra obtained

with the OFCEAS system for a temperature of 50°C and pressure of 5, 15 and 20 mbar. The increase of the baseline level and of the absorption lines is clearly apparent. The baseline is not completely flat because of optical interference pattern which is produced between the output cavity mirror and the photodiode. Several efforts have been made in order to minimize




these effects, but it was not possible to remove them completely. However, these optical fringes are taken into account in the spectral fit and therefore removed for the spectral analysis.

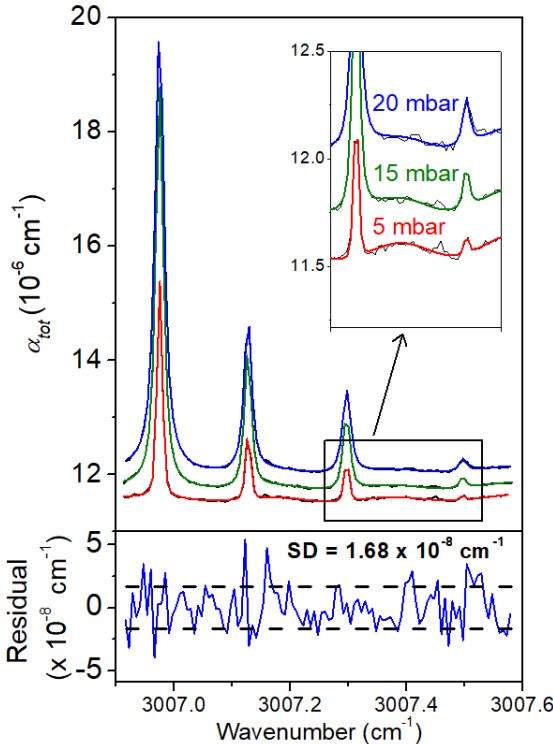

**Figure 1:** *Upper panel:* Total absorption coefficient measured by OFCEAS when the optical cavity is filled by water at three different pressures for a temperature of 50 °C. Black lines are experimental data and color lines the corresponding spectral fit.
*Lower panel*: Residuals of the fit for the spectrum at 20 mbar.

## 2.2. Self-continuum cross-section retrieval

The OFCEAS absorption coefficient is the sum of the contribution of the losses of the (evacuated) cavity, $\alpha_{cavity}$, and of water absorption. The latter includes water vapor monomer local absorption lines (WML) and water vapor self-continuum (WCS):

$$\alpha_{tot}(\nu, T) = \alpha_{cavity}(\nu) + \alpha_{WML}(\nu, T) + \alpha_{WCS}(\nu, T) \tag{1}$$

Note that Rayleigh scattering contribution is negligible (Thalman et al., 2014).

It is worth noting that $\alpha_{WML}$ accounts not only for the four water lines observed in the spectra (**Fig. 1**) but also for the contribution of the wings of the lines located outside the recorded region. As the recorded spectral interval corresponds to a





micro window surrounded by lines with intensity up to 4-5 orders of magnitude larger than those observed in **Fig. 1**, the monomer wing contribution is in fact the dominant contribution to the base line increase. According to the usual definition of the water continuum, lines located in the $[\nu-25, \nu+25 \text{ cm}^{-1}]$ wavenumber interval have to be considered and simulated using the HITRAN2016 database with a Voigt profile truncated at $\pm25 \text{ cm}^{-1}$ from the line center. Note that similarly to the

continuum, the monomer wing contribution has quadratic pressure dependence.

For each spectrum, the baseline is determined by subtracting the four absorption lines. A Rautian profile was used to fit the line profile by fixing the Doppler widths to the theoretical values and fitting the Lorentzian collisional broadening and Dicke narrowing for each individual absorption line. The baseline is fitted by a third order polynomial in the basis of Legendre Gauss polynomials. Optical fringes due to parasite reflections which induce etaloning effects are taken into account by

adding three sine waves at fixed frequency and amplitude with free phase (frequencies and amplitudes were determine by FFT analysis of the residual). The quality of the spectrum simulation is illustrated by the residuals included in the bottom panel of **Fig. 1**, showing an *rms* of the residuals of $1.68\times10^{-8} \text{ cm}^{-1}$ at 20 mbar for a single scan (100 ms). As mentioned above, the pressure value of each spectrum was calculated using the water absorption lines. The integrated absorption coefficient of a given line is simply the product of the molecular density (in $\text{molecule.cm}^{-3}$) by the line intensity (in

cm/molecule) as tabulated in the HITRAN2016 database. In the studied interval, the two strongest lines at 3006.975 and 3007.1278 $\text{cm}^{-1}$ are due to the main isotopologue ($H_2^{16}O$). Their intensities were recently measured by Loos et al. (2017) with an accuracy of 1.5 % and 2.4 %, respectively. As a validation test, we present in **Fig. 2** the ratios of the pressure values derived from these two lines during a pressure ramp at 50 °C. The small dispersion of the ratio values along the ramp confirms the stability of the experiment (average value of 1.056 with a standard deviation of 1.0 %). Considering all the

ramps recorded at different temperatures, we obtain an average ratio of 1.041 and a standard deviation of 0.84 %, which is consistent with the intensity error bars reported by Loos et al. (2017). Note that compared to Loos et al. (2017), the *ab initio* values of the BT2 (Barber et al., 2006) and Schwenke and Partridge (2000) theoretical databases of water vapor transitions gives ratio of 1.048 and 1.061, respectively. As a result, the pressure of each spectrum was obtained as the average value of the two determinations with an associated error bar estimated to 3 %.







**Figure 2:** *Upper panel:* Ratio of the pressure values deduced from the $H_2^{16}O$ line at 3006.975 cm$^{-1}$ ($P_1$) and at 3007.127 cm$^{-1}$ ($P_2$) as a function of $P_1$ for a pressure ramp at 50 °C

*Lower panel:* Corresponding variation of the water continuum absorption near 3007 cm$^{-1}$ as a function of the squared pressure. The plotted
5   data correspond to the superposition of 5 distinct pressure ramps after smoothing ten adjacent measurements points. The solid black line corresponds to the best linear fit. The constant term due to the cavity losses obtained from the fit was subtracted. The contributions of water monomer wings and of the self-continuum are highlighted (in grey and cyan, respectively).

**Fig. 2** presents the dependence of the zero order term of the baseline fit *versus* the squared pressure, during a pressure ramp

10   at 50°C. A convincing quadratic dependence is obtained. After subtraction of the constant term due to the cavity losses, the

plotted quantity results as the sum of the water vapor continuum $(\alpha_{WCS})$ and the monomer wing contributions, which have

both a quadratic pressure dependence. Let us note that the wing contribution, calculated as indicated above, is more than

twice the water vapor continuum. The large number of scans performed for each pressure ramp (typically, 6000 spectra per

ramp of 10 min duration) allows increasing the precision of the fitted value of the quadratic coefficient, $\alpha_{WCS}(\nu, T)/P^2$

15   which defines the self-continuum cross-section, $C_S$, (in cm$^2$ molecule$^{-1}$ atm$^{-1}$): $\alpha_{WCS}(\nu, T) = \dfrac{1}{k_B T} C_S(\nu, T) P^2$, where $k_B$ is

the Boltzmann constant. The obtained $C_S$ values are listed in **Table 1** for six temperature values ranging from 25°C to 50 °C



**Table 1.**
Self-continuum absorption cross-sections of water vapor measured near 3007 and 5000 cm$^{-1}$

| Wavenumber (cm$^{-1}$) | T(K) | $C_S$ (10$^{-23}$ cm$^2$ molecule$^{-1}$ atm$^{-1}$) |
|---|---|---|
| 3007 | 298.15 | 1.96(39) |
| | 303.15 | 2.17(54) |
| | 308.15 | 1.83(46) |
| | 313.15 | 1.43(36) |
| | 318.15 | 1.82(46) |
| | 323.15 | 1.40(35) |
| 4995.63 | 297.82 | 0.757(38) |
| 4998.98 | 297.48 | 0.776(39) |
| 5002.05 | 297.80 | 0.798(40) |
| 5006.67 | 297.32 | 0.794(40) |

*Note*

*Numbers in parenthesis correspond to the 1-σ overall estimated error bars in units of the last quoted digit.*

In the present case, the error bars on $C_s$ values are relatively large, on the order of 30 %. This results from different factors. The statistical error bars of the fitted value of the quadratic coefficient is on the order of 5 %, mainly due to fluctuations of the spectra baseline level related to the short ring down time provided by the cavity mirrors. Taking into account that the

water continuum represents about 25 % of the measured signal (**Fig. 2**), this fit error gives a 15-20 % relative uncertainty on $C_s$. The uncertainty on the pressure values derived from the absorption lines contributes also to the error budget (see above). It results from the uncertainty on the intensity values reported by Loos et al. (2017) and from the ±1 K uncertainty on the gas temperature that affects line intensities (the two water lines have a relatively high lower state energy - 1813.8 and 1394.8 cm$^{-1}$ - leading to a strong temperature dependence of their intensities i.e. 2.3% per K for the strongest line). In Richard et al.

(2017), a pressure ramp of dry nitrogen up to 40 mbar was performed, proving that the optical signal is not sensitive to pressure changes. Thus the cavity instability is not expected to contribute significantly to the error budget. Overall, we estimate the 1-σ error bar of the $C_S$ values to be 20 % for the room temperature value (298 K) and 25 % at higher temperature.

**2.3. Comparison to literature**

The different experimental determinations of the self-continuum cross-sections at room temperature in the 4 µm window are gathered in **Fig. 3** together with some of the last versions of the MT_CKD continuum up to the current V3.2 version. In addition to the present study at 3007 cm$^{-1}$ located at the high energy edge of the window, previous OFCEAS studies were performed at 2283 cm$^{-1}$ (Campargue et al., 2016) and 2491 cm$^{-1}$ (Richard et al., 2017). Three datasets were obtained by FTS with long-path absorption cells and cover most of the transparency window: *(i)* CAVIAR consortium dataset obtained with a

512.7 m pathlength at 293 K (Ptashnik et al., 2011a), *(ii)* two series of FTS determinations from IAO in Tomsk (Ptashnik et

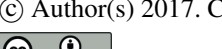



al., 2013; 2015), obtained with a pathlength of 612 m at 289.5 and 287 K, respectively, *(iii)* the FTS dataset obtained at NIST by Baranov and Lafferty (2011) with a 100 m pathlength between 311 and 363 K. Finally, Burch and Alt (1984) reported continuum measurements in the center of the window using a grating spectrograph and an optical configuration aiming to minimize the spectra baseline drifts.

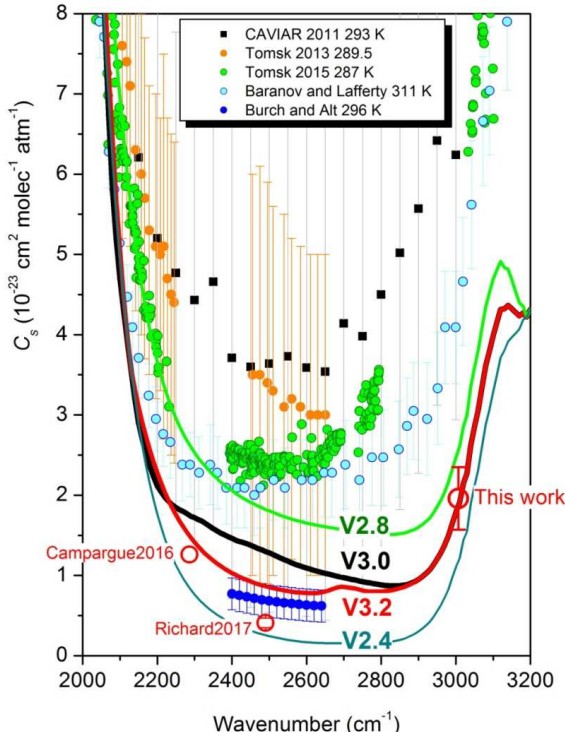

**Figure 3:** Overview comparison of the self-continuum cross-section of water vapor at room temperature.
Solid lines show different versions of the MT_CKD model. Experimental results are obtained using different techniques, namely by OFCEAS, (red circles; Campargue et al, 2016; Richard et al., 2017; this work); by FTS from Baranov and Lafferty (2011), light blue circles, from CAVIAR (black squares; Ptashnik et al., 2011a), from Tomsk2013 (orange circles; Ptashnik et al., 2013), from Tomsk2015 (orange circles; Ptashnik et al., 2015) and with a grating spectrograph by Burch and Alt (1984), dark blue circles. Note that the 30-50 % error bars on Tomsk2015 values are not plotted for clarity.

At 3007 cm$^{-1}$, the present OFCEAS and the MT_CKD3.2 values are in excellent agreement. The comparison to the FTS values is similar to that encountered in most of the transparency windows: the FTS values are largely overestimated compared to both MT_CKD and the laser measurements. Compared to OFCEAS, CAVIAR and Tomsk 2015 $C_S$ values, reported with a 50 % error bar, are overestimated by a factor of three while the FTS values reported by Baranov and Lafferty with a 22% error bar are overestimated by a factor of two.



The evolution of the MT_CKD model in the 4.0 µm over the last ten years is illustrated by the 2.4, 2.8, 3.0 and 3.2 versions plotted in **Fig. 3**. In the center of the window, the different versions have varied over one order of magnitude amplitude. On the basis of the OFCEAS measurement at 2283 cm$^{-1}$ (Campargue et al. 2016), the MT_CKD3.2 version of the self-continuum has been recently decreased in the low energy edge of the window (the very recent OFCEAS measurement at

2491 cm$^{-1}$ (Richard et al., 2017) is not yet taken into account). In the high energy edge of the window, the present measurement at 3007 cm$^{-1}$ validates the V3.2 values which are unchanged from V3.0.

**3. CRDS near 5000 cm$^{-1}$**

*3.1 Spectra acquisition and self-continuum cross-section retrieval*

The reader is referred to (Čermák et al., 2016; Campargue et al. 2017) for a description of our CRDS spectrometer dedicated

to the high sensitivity spectroscopy of atmospheric species in the 2.1 µm window. The setup is an improved version of that used for absorption lines and continuum measurements of water vapor (Campargue et al. 2017) and $CO_2$ (Mondelain et al., 2017) absorption. A Distributed Feed Back (DFB) diode laser (from Eblana Photonics) covering the 4995-5007 cm$^{-1}$ spectral interval was used as light source. The optical feedback technique (Lin et al. 2012) was used to narrow the laser emission line width and increase the light injection into the cavity during resonances with cavity modes. The frequency of the laser diode

was measured with a wavelength meter (model 621-A IR from Bristol Instruments, 8 MHz 1-σ accuracy and 2.5 Hz acquisition rate). The present measurement points of the water continuum near 5000 cm$^{-1}$ are located on the high energy edge of the 2.1 µm window and complete our spectral sampling of the window by OFCEAS and CRDS between 4250 and 4725 cm$^{-1}$.

**Fig. 4** shows an overview of the water vapor CRDS spectrum near 5002 cm$^{-1}$ and the location of one of the four spectral

points selected for the water vapor absorption continuum retrieval. From the spectrum recorded with an unprecedented sensitivity in the region ($\alpha_{min}$~order of $3\times10^{-10}$ cm$^{-1}$), water vapor transitions with intensity values below $1\times10^{-28}$ cm/molecule are measured. The recorded spectrum was found in very good agreement with simulations performed on the basis of the HITRAN2016 database (see **Fig. 4**).



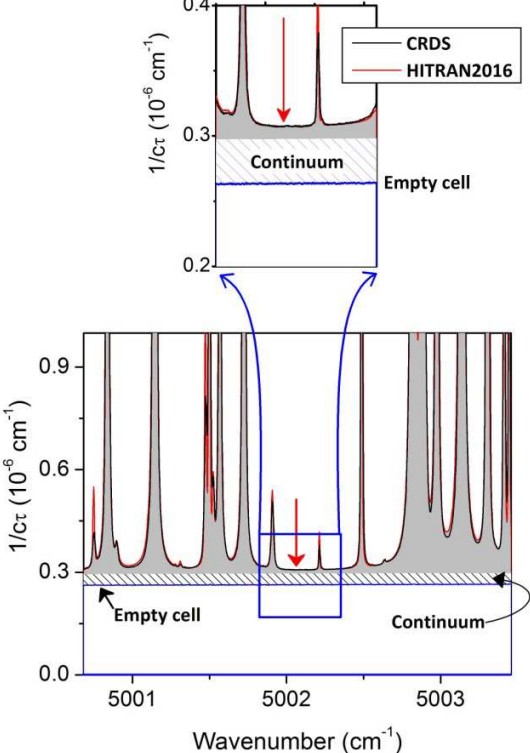

**Figure 4:** CRDS spectrum of water vapor between 5000.6 and 5003.5 cm$^{-1}$. The pressure was 7.0 Torr at 297.35 K. The red arrow indicates one of the four spectral points located in an interval free of absorption lines, which were chosen for the retrieval of the water vapor absorption continuum. The simulation performed on the basis of the HITRAN2016 database (red solid line) is used to subtract the monomer contribution. The water continuum (dashed area) corresponds to the gap between the spectrum with the monomer simulation subtracted and the baseline level recorded with the evacuated cell.

The CRDS cell was filled with water vapor after purification by liquid nitrogen cooling and pumping on the residual vapor phase. The pressure in the cell was continuously monitored with a capacitive transducer (model ATM.1ST from STS; 50 mbar full range; accuracy of ±0.1 % of the full range). Contrary to the above OFCEAS measurements, for which the self-continuum was derived from a series of spectra, the present CRDS measurement was performed at fixed frequency by monitoring the variation of the cavity loss rate, $1/c\tau$, during pressure ramps. The typical value of the ring-down time of the evacuated cavity, $\tau_0$, was about 124 μs and the acquisition rate was about 60 ring-down events per second. About 8000 ring-down events were measured during upward and downward pressure ramps up to 15 Torr, at four selected spectral points free of absorption lines (see Table 1). The typical duration of the ramps was 90 s and the cell temperature was about 297.5 K.





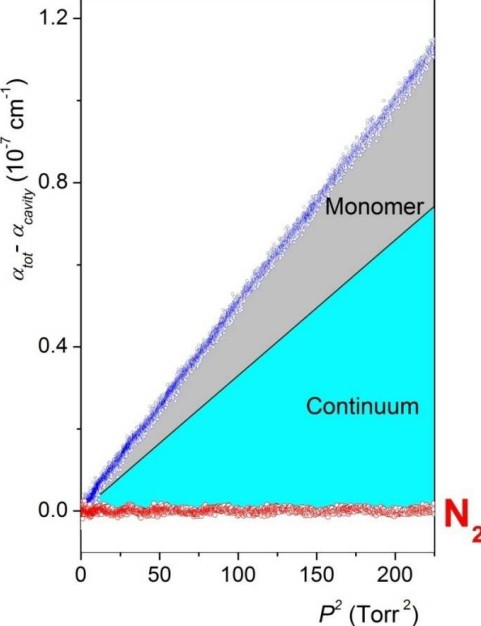

**Figure 5:** Variation of the loss rate measured by CRDS at 4995.63 cm⁻¹ during the filling of the cell with water vapor up to 15 Torr. The constant term due to the cavity losses obtained from the fit was subtracted. Every blue dot corresponds to a single ring-down event. Red dots represent similar measurements performed with pure nitrogen. The contributions of water monomer and of the self-continuum are highlighted (in grey and cyan, respectively).

Fig. 5 shows the variation of the loss rate at 4995.63 cm⁻¹ during the filling of the CRDS cell. No deviation from the quadratic law is noted up to the maximum pressure value of 15 Torr corresponding to 65 % humidity rate. The loss rate is the sum of the loss rate with the cell evacuated and of the absorption coefficient, $\alpha(\nu)$, due to water vapor:

$$\frac{1}{c\,\tau(\nu)} = \frac{1}{c\,\tau_0(\nu)} + \alpha(\nu) \tag{2}$$

where $c$ is the speed of light. The Rayleigh scattering contribution being negligible, the absorption coefficient includes contributions due to water vapor monomer local lines (WML) and water vapor self-continuum (WCS). The water absorption continuum was then obtained after subtraction of the calculated WML contribution calculated as indicated above. According to the spectral points, the WML contribution represents between 18 and 38 % of the total absorption (see **Fig. 5**). The self-continuum absorption cross-sections included in **Table 1** were derived from the linear fit of the absorption continuum in $P$ squared. As a result of the large number of measurements for each pressure ramp and of the very good mechanical stability of the cavity, the statistical error on the fitted coefficient is very small, less than 1 %. The mechanical and optical stability of





the spectrometer was checked by using nitrogen ($N_2$) instead of water vapor. The results included in **Fig. 5** confirm that the measurements are not affected by the pressure variation during the pressure ramp.

The $C_S$ values listed in **Table 1** are values averaged for an upward and downward pressure ramps. Maximum differences of less than 2 % were obtained between upward and downward determinations. Taking into account other error sources (0.25 % on the pressure measurements, 0.2 °C on the temperature, fit error) we estimate to less than 5 % the relative uncertainty on the listed $C_S$ values.

**3.2. Comparison to literature**

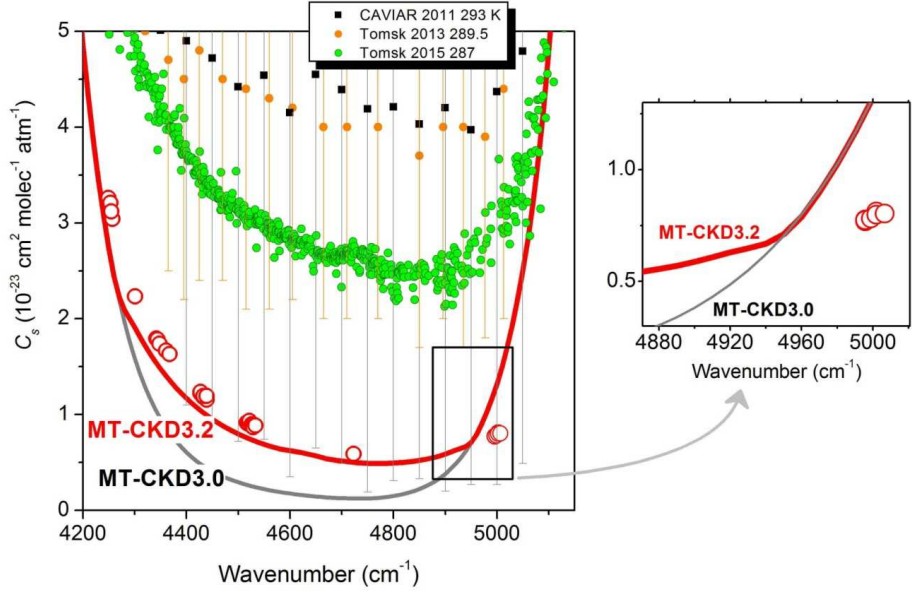

**Figure 6:** Spectral dependence of self-continuum cross-section at room temperature in the 2.1 µm window. The two last versions of the MT_CKD model (V3.0 and V3.2, black and red solid lines, respectively) are compared to the laser based measurements (red circles), to the FTS results obtained in Tomsk (Ptashnik et al., 2013, 2015; green and orange circles) or by the CAVIAR consortium (Ptashnik et al., 2011a) (black squares). The 30-50 % error bars on the Tomsk2015 FTS values and the small error bars on our laser based values are not plotted for clarity. The zoom highlights the present CRDS values near 5000 cm$^{-1}$.

Although the four spectral points presently selected span less than 12 cm$^{-1}$, a significant increase is observed with the frequency for the self-continuum cross-sections presented in the inset of **Fig. 6. Fig. 6** shows an overview comparison of previous FTS and laser based measurements in the 2.1 µm window with the V3.0 and V3.2 versions of the MT_CKD model. On the basis of our preceding CRDS and OFCEAS measurements below 4750 cm$^{-1}$, the very recent V3.2 version was significantly increased in the 4250-4950 cm$^{-1}$ range. Above this value, the V3.0 version was kept unchanged. The present $C_S$



values near 5000 cm$^{-1}$ are fully consistent with our previous results and on line with a linear extrapolation of the V3.2 model above 4950 cm$^{-1}$. It indicates that the present MT_CKD spectral dependence is validated up to 4950 cm$^{-1}$ but should be decreased by about a factor of 2 at 5000 cm$^{-1}$. Similarly to the situation found in the 4.0 µm, the FTS values of the CAVIAR consortium and of the two Tomsk datasets are largely overestimated, by about a factor of 5 in the present case. In our

opinion, as the three sets of FTS values are reported with very large error bars (up to ±94 % for the CAVIAR value), they should be considered as upper limit values corresponding to the detectivity threshold of the FTS setups.

## 4. Temperature dependence

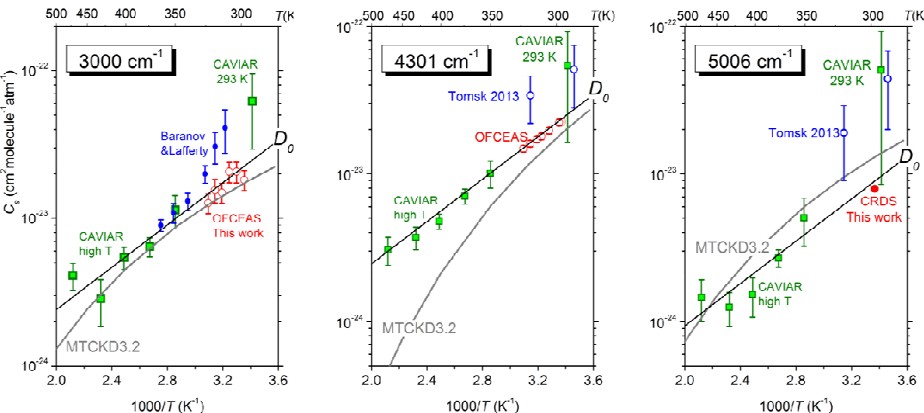

**Figure 7:** Temperature dependence of the water vapor self-continuum cross-sections near 3000, 4301 and 5006 cm$^{-1}$ obtained by OFCEAS
and CRDS (open and full red circles, respectively), by FTS (green squares: CAVIAR; full blue circles: Baranov and Lafferty (2011); open blue squares: Tomsk 2013). The MT_CKD3.2 values which are normalized to the number density at 1 atm and 296 K were multiplied by 296/$T$. The $D_0$ slope corresponds to an exp($D_0$/k$T$) law, $D_0 \approx 1100$ cm$^{-1}$ being the dissociation energy of the water dimer molecule.

The characterization of the temperature dependence of the water vapor self-continuum is an important issue in particular in
the temperature range relevant for atmospheric applications. While the water continuum is stronger at low temperature, it is in fact easier to measure it at high temperature where the pressure limitation due to the saturation pressure is strongly relaxed. We have collected in **Fig. 7,** the self-continuum cross section values provided in the literature at different temperatures around our measurement points at 3007 and 5000 cm$^{-1}$ and plotted them in logarithmic scale *versus* 1/$T$. The present OFCEAS measurements cover the 298-323 K interval and shows a decreasing tendency with temperature. In
addition to the room temperature FTS studies discussed above, prior measurements at 3000 cm$^{-1}$ were reported by the CAVIAR consortium at temperatures from 350 to 472 K using a short-path absorption cell and vapor pressure values up to





1.6 atm while Baranov and Lafferty (2011) reported FTS measurements in the 311-363 K range. We have also included in **Fig. 7**, a similar plot for the 4301 cm⁻¹ spectral point measured by OFCEAS (Ventrillard et al.; 2015). At 4301 cm⁻¹, the whole set of measurements follows a simple exponential law in $1/T$ with a slope close to the dissociation energy of the water dimer, $D_0 \approx 1100$ cm⁻¹ (Rocher-Casterline et al., 2011). The set of measurements at 3000 cm⁻¹ is consistent with the same

$\exp\left(\dfrac{D_0}{kT}\right)$ temperature dependence but less convincing as a result of larger error bars on the OFCEAS and CAVIAR high

temperature values. The situation is mostly similar at 5006 cm⁻¹ where our accurate $C_S$ value at room temperature is on line with the high temperature CAVIAR values reported with larger error bars.

Overall, the MT_CKD temperature dependence at 3007 and 5000 cm⁻¹ agree with the available experimental data (see **Fig. 7**). At 4301 cm⁻¹, the MT_CKD temperature dependence is stronger in particular for high temperature values.

Nevertheless, near room temperature, the MT_CKD dependence is close to the $\exp\left(\dfrac{D_0}{kT}\right)$ law at the three considered spectral

points.

## 5. Conclusion

Using cavity-enhanced absorption methods, the infrared water vapor continuum has been determined at two new frquencies of the 4.0 and 2.1 µm windows. We present in **Fig. 8** an exhaustive comparison of the experimental determinations of self-

continuum cross-sections to the MT_CKD3.0 and 3.2 models from 1500 to 9000 cm⁻¹. As mentioned above, the V3.2 version has been adjusted according to part of the laser based studies listed in **Table 2**. In the 1.6 and 1.25 µm windows where the continuum is very weak and where previous experimental results were practically absent, the CRDS measurements were found in an overall good agreement with previous versions of the MT_CKD model and the V3.2 adjustments are small in the center of the windows. The largest correction concerns the low energy edge of the 1.25 µm window near 7500 cm⁻¹ where

the MT_CKD continuum was decreased by a factor of 3. Surprisingly, more significant corrections have concerned the 4.0 and 2.1 µm windows which have a larger atmospheric impact. Compared to V3.0, the V3.2 $C_S$ values were decreased by about 40 % near the center of the 4.0 µm window. Nevertheless, our recent OFCEAS measurement at 2491 cm⁻¹ (not taken into account in V3.2) indicates that the V3.2 continuum is still overestimated by a factor of 2 in the center of the window while the present results at 3307 cm⁻¹ validate the high energy edge of the 4.0 µm window within a 20 % error bar. Some

additional measurement points are still desirable to complete the coverage of the 4.0 µm window. The 2.1 µm window has been sampled by CRDS and OFCEAS at seven spectral intervals. This very consistent set of measurements (**Fig. 6**) has led to a strong increase of the continuum in this window (a factor of 4 near the center). At the high energy edge, above 4950 cm⁻¹, the V3.0 values were kept unchanged and an overestimation by about a factor of 2 is noted compared to the present CRDS results near 5000 cm⁻¹. As illustrated in **Fig. 6**, the V3.2 correction should be extended at higher energy resulting in a wider

2.1 µm window.



The strong disagreement with FTS measurements (Ptashnik et al., 2011a, 2013, 2015) has been discussed by Shine et al., 2016 and Campargue et al. 2016. The measurement of a very small variation of a FTS spectrum baseline induced by the injection of water vapor in an absorption cell requires very high baseline stability during the entire measurement period. In our opinion, traditional methods such as FTS with absorption pathlengths limited to a few hundred meters do not provide

sufficient sensitivity and stability to evaluate sub % baseline variations (For instance, in the center of the 4.0 and 2.1 µm windows, at 15 mbar, the continuum signal to be measured leads to a 0.1-0.2 % light attenuation for an 500 m absorption pathlength). As underlined by Burch and Alt (1982, 1984, 1985), experimental biases related to optical or mechanical changes lead generally to overestimated values of the continuum. We believe that the high and mostly constant Tomsk and CAVIAR $C_s$ values of about $4\times10^{-23}$ cm$^2$molec$^{-1}$atm$^{-1}$ reported for the 4.0, 2.1 and 1.6 µm windows should be considered as

an upper limit due to the detectivity threshold of the FTS approach. value. In general, the retrieval of a very weak water continuum from a single pressure spectrum is hazardous. The squared pressure dependence of the absorption signal as systematically fulfilled in the CRDS and OFCEAS studies provides a crucial test to insure that baseline variation of the spectra has a gas phase origin. This is an important advantage of the laser based approach to allow monitoring the pressure dependence of the continuum absorption during pressure ramps lasting no more than a few minutes. In addition, using

similar pressure ramps with non-absorbing gases ($N_2$ or Ar), the baseline stability of the laser setups can easily be quantitatively evaluated and was found to be very high in the range of pressure used.

**Table 2**
Summary of the self-continuum retrievals of water vapor performed by CRDS and OFCEAS

| Window Approx. center | Measurement points (cm$^{-1}$) | Method | Laser source [a] | T (K) | Ref. |
|---|---|---|---|---|---|
| 2500 cm$^{-1}$ (4.0 µm) | 2283 | OFCEAS | QCL | 296, 301, 312 | Campargue et al., 2016 |
| | 2491 | OFCEAS | ICL | 297, 312, 325 | Richard et al., 2017 |
| | **3007** | **OFCEAS** | **ICL** | **298-323** | **This work** |
| 4700 cm$^{-1}$ (2.1 µm) | 4249-4257 (4 pts) | CRDS | DFB | 298 | Mondelain et al., 2015 |
| | 4301 | OFCEAS | DFB | 297-323 | Ventrillard et al., 2015 |
| | 4341-4367 (5 pts) | CRDS | VECSEL | 297 | Campargue et al., 2016 |
| | 4427-4441 (3 pts) | CRDS | DFB | 296 | Richard et al., 2017 |
| | 4516-4532 (8 pts) | CRDS | DFB | 297 | Campargue et al., 2016 |
| | 4723 | OFCEAS | DFB | 298-323 | Ventrillard et al., 2015 |
| | **4995-5007 (4 pts)** | **CRDS** | **DFB** | **297.7** | **This work** |
| 6300 cm$^{-1}$ (1.6 µm) | 5875-6665 (10 pts) | CRDS | DFB | 302-340 | Mondelain et al. 2013; 2014 |
| 8000 cm$^{-1}$ (1.25 µm) | 7500-7920 (30 pts) | CRDS | DFB | 297 | Campargue et al., 2016 |
| | 7920-8300 (26 pts) | CRDS | ECDL | 297 | |

*Note.*
[a] QCL: Quantum Cascade Laser, ICL: Interband Cascade Laser, DFB: Distributed Feed Back diode laser, VECSEL: Vertical External Cavity Surface Emitting Laser.


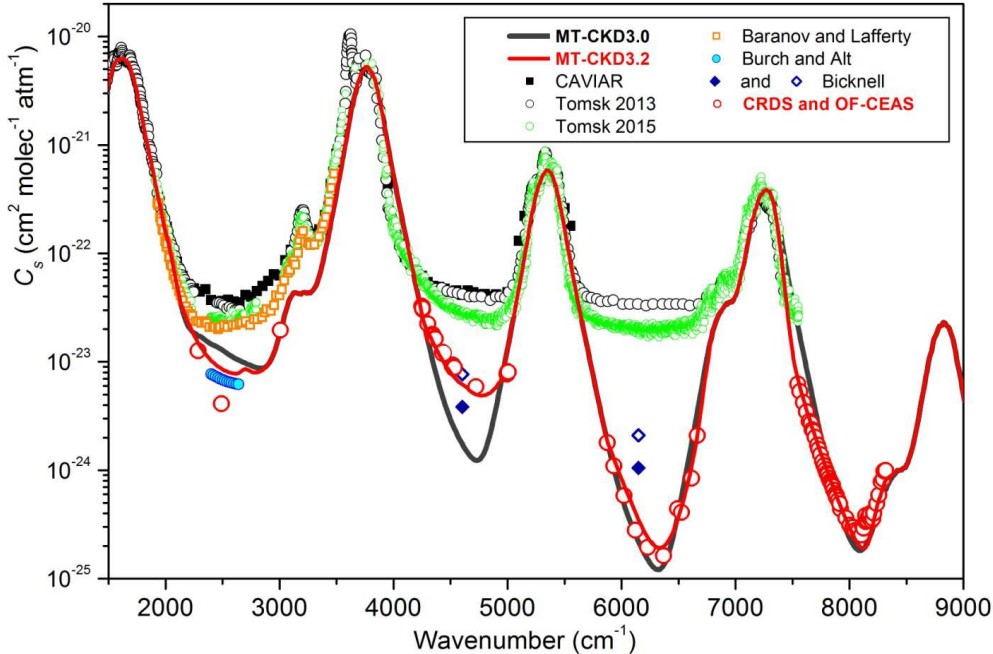

**Fig. 8**
Comparison of the MT_CKD3.0 and 3.2 models (black and red solid lines, respectively) (Mlawer et al. 2012 ) of the water vapor self-continuum cross-sections, $C_S$, in the 1500-9000 cm$^{-1}$ range to an exhaustive collection of the experimental determinations: *(i)* FTS values reported by Baranov and Lafferty (2011) (orange squares); the CAVIAR consortium (Ptashnik et al., 2011a) (black full squares), from
5 Tomsk2013 and Tomsk2015 experiments (Ptashnik et al., 2013, 2015) (black and green open circles, respectively); *(ii)* results by Bicknell et al. (2006) from calorimetric-interferometry in air at 4605 cm$^{-1}$ (blue open diamond corresponds to a measurement in air, blue full diamond is an estimation of the self-continuum contribution) *(iii)* measurements by Burch and Alt (1984) near 2500 cm$^{-1}$ using a grating spectrograph (open blue circles); *(iv)* present and previous measurements by CRDS and OFCEAS (red open circles).

Much laboratory work is still required to characterize the water continuum and fulfill the needs for atmospheric applications. In particular, only very partial information is available concerning the temperature dependence. By gathering OFCEAS results near room temperature with previous high temperature FTS measurements, the temperature dependence at several spectral points of the 4.0 and 2.1 µm windows was found to follow a simple exp($D_0/kT$) law, $D_0$ being the dissociation
15 energy of the water dimer (~1100 cm$^{-1}$) (Ventrillard et al., 2015; Richard et al. 2017). Let us note that partly resolved absorption features due to water dimers have been identified in the mm-range room-temperature spectrum of water vapor (Tretyakov et al., 2013) and evidence of the water dimer contribution to the continuum was reported in the 14-35 cm$^{-1}$ region (Odintsova et al. 2017) and in the region of the absorption bands (Ptashnik et al., 2008; 2011b). Nevertheless, in absence of





theoretical support, it seems hazardous to consider the $\exp(D_0/kT)$ empirical law as an experimental signature of the water dimer origin of the continuum in the windows.

In the Earth's atmosphere, the foreign-continuum due to the interaction of water molecules with nitrogen and oxygen may have a contribution comparable to that of the self-continuum (Mondelain et al. 2015). Concordant experimental results

(Ptashnik et al., 2012) have led to a significant increase of the MT_CKD foreign-continuum in the windows but measurements are scarce and should be extended in particular near room temperature. Note that the quality of the foreign-continuum retrieval from spectra of water in air depends on the accuracy of the self continuum contribution which has to be subtracted. We have demonstrated that cavity enhanced laser absorption methods provide a powerful approach for an accurate characterization of the self continuum near room temperature. In the future, we plan to adapt our experimental

setups to perform similar studies of the water vapor foreign-continuum.

In recent works, Paynter and Ramaswamy (2011, 2012) constructed an empirical water vapor continuum (BPS-MTCKD continuum) by modifying the MT_CKD model in the near infrared windows mostly according to FTS values of the CAVIAR consortium. They showed that the self- and foreign-water vapor continuum could result in between 1.1 and 3.2 $Wm^{-2}$ additional clear-sky absorption of solar radiation, with this large error bar being due to the "fairly large measurement

uncertainties in the shortwave near-infrared window". Ptashnik et al. (2011a) also quantified the impact of the CAVIAR self-continuum cross-sections upon solar radiation absorbed by the atmosphere and obtained a significant additional global mean absorption of ~ 0.75 $Wm^{-2}$ relative to the MT_CKD2.5 i.e. about 1% of the total clear-sky absorption. During the very recent years, a set of CRDS and OFCEAS self-continuum cross-sections were obtained over four transparency windows with accuracy as good as a few %. We hope that these results and the forthcoming studies of the foreign-continuum will help to

reduce significantly the uncertainty of the impact of the water continuum on the Earth's clear-sky radiation budget.

**Acknowledgements**

This project is supported by the LabexOSUG@2020 (ANR10 LABX56)and the LEFE-ChAt program from CNRS-INSU. The research leading to the results at 3.33μm has received funding from the European Community's Seventh Framework Program ERC-2011-AdG under grant agreement n° 291062 (ERC ICE&LASERS), as well as ERC-2015-PoC under grant

agreement n° 713619 (ERC OCEAN-IDs).



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
