# Peer review of "The water vapor self-continuum absorption in the infrared atmospheric windows"

_Atmospheric Measurement Techniques, 2017_

## Referee Comment (RC1) · Anonymous Referee #1 · 9 Jan 2018

The manuscript "The water vapor self-continuum absorption in the infrared atmospheric windows: New laser measurements near 3.3 and 2.0 microns" by Lechevallier et al. presents the latest water vapor self continuum measurements from the Grenoble group, which has been steadily advancing in their quest to perform high-quality self continuum measurements in all near-IR windows of consequence to global energy balance. This new paper analyzes measured values at the high energy end of two windows, regions in which no measurements of equivalent accuracy have been performed previously. The authors ably present the measurement approach employed and compare their results to previous measurements and the MT_CKD model. The analysis is solid and the paper will make a useful addition to the literature on this subject. Acceptance is

recommended, although the authors should implement the improvements suggested below, as well as improve some instances of awkward language (further below).

Issues:

1) The uncertainty in the derived self continuum values due to uncertainty in line widths of neighboring strong lines does not seem to be considered. The analysis presented in both regions indicates that the line contribution to the absorption coefficient is significant – for the 3007 cm-1 line it is even larger than the self continuum contribution. Therefore, any substantial uncertainty in the self-broadened line widths would lead to uncertainty in the derived self continuum coefficients. Unfortunately accurate information about the self-broadened line width uncertainty is hard to come by – the best approach might be to assume that it's similar to the uncertainty in the foreign-broadened widths. To develop the aer_v3.6 line parameter file, high-resolution TCCON observations in the near-IR were used to analyze and, if necessary, modify the foreign widths from three sources: HITRAN 2012, Mikhailenko et al., and one from the Regalia group. The differences between the three compilations and widths modified for aer_v3.6 could be as much as 20%. Therefore, for the present manuscript, I recommend the uncertainty provided for the derived self continuum coefficients should be revised to reflect a self line width uncertainty of ∼20%.

2. This paper states (pg. 10, line 5 and pg. 15, line 22) that MT_CKD has not yet taken the Richard et al. measurement into account. That leaves the reader with the wrong impression. The Richard et al. value was considered in developing recent MT_CKD version, but found to not be consistent with the satellite- and ground-based observations analyzed in Mlawer et al. (2012). Therefore MT_CKD was reduced as much as possible in that region to reflect the existence of the Richard et al. value but still maintain agreement with these field observations.

This paper does a thorough job comparing the new measurements and the previous ones from the Campargue group with prior ones from other teams, but puzzlingly does

not present MT_CKD as being measurement-based in this window. This leaves the reader with a misleading impression. The disagreement between the field observations analyzed in Mlawer et al. and the Richard et al. self continuum measurement will hopefully be resolved based on further observations, such as the foreign continuum measurements planned by the Grenoble team (the foreign continuum impacts the analysis of the field measurements) and additional self measurements in this window.

3. On page 18, last paragraph, the authors have a fair amount of text discussing various analyses of atmospheric absorption based on the FTS measurements. Since this paper and the others from the Grenoble group have basically dismantled the possibility of the self continuum in the near-IR windows being as high as the room-temperature FTS measurements indicated (i.e. significantly larger than MT_CKD), these high estimates of absorptions can be ruled out. Reading this last paragraph, I expected to have this paper culminate in a clear statement that these high estimates can now be assumed to have been in error. Is there a reason why the authors avoid stating this conclusion?

Typos, minor items:

pg1 line 11 – "in the recent years" should be "in recent years" line 19 – MT_CKD3 should be MT_CKD

pg2 line 2 – A fairly small fraction of the solar radiation in the earth's atmosphere is absorbed by water vapor (or absorbed at all). The authors likely mean that 60% of the solar radiation that is absorbed is absorbed by water vapor. line 3 – Since there are all sorts of water vapor absorption bands moving into the visible from the infrared, with weaker vibrational bands spaced with not perfect regularity, the authors might want to qualify "located every 1500-2000 cm-1" with a word like ""roughly" or "more-or-less". line 4 – Suggested change for clarity: "These are formed by multitudes of narrow rovibrational absorption lines that are...." lines 18-19 - Suggested change: "This model, which is basically a far-wing line shape model in the window regions, involves a number

of ad hoc parameters that have been..." line 20 – "long-infrared" is not typically used. line 22 – The comma after "recently" should probably be removed.

pg3 line 2 – The word "windows" should follow "4.0, 2.1, 1.6 and .25 microns" line 7 – "baseline" is usually one word line 21 – A period is missing at the end of the sentence.

pg 4 line 21 – "upstream of" line 31 – Add "with pressure" after "absorption lines".

pg 5 line 5 – The reader will have no idea when reading the caption to Figure 1 what is meant by "spectral fit" since its meaning is only clear (the absorption coefficient calculation) when reading about the calculation on page 6 – for example, they might assume it refers to some sort of curve fitting from the data. Perhaps change it to "corresponding spectral fit, as described in the text" or "corresponding absorption coefficient calculation, as described in the text".

pg 9 line 15 – For clarity, add the word "the" between "OFCEAS" and "CAVIAR".

pg 10 lines 17-19 – Since 5000 cm-1 is not between 4250 and 4725 cm-1, the wording of this sentence should be changed.

pg 15 line 10 – The use of "law" here implies that the exponential expression is some sort of physical law that the values plotted in Figure 7 should obey. A word like "expression" might be more consistent with the intended meaning. line 13-14 – "new frquencies of" should be "new frequencies in" (note the missing letter "e") line 15 - V3.2 version is redundant since "V" stands for "version"

pg 18 line 5 – It's unclear why there is a reference to Ptashnik et al. here. According to the MT_CKD web page, the recent modifications to the MT_CKD foreign continuum in the near-IR are due to "Foreign continuum coefficients from 1800-3000 cm-1 were modified to improve agreement with Baranov and Lafferty (2012); in the 1900-2150 cm-1 region, attention was also paid to IASI measurements (Alvarado et al., 2013).... Foreign continuum coefficients at wavenumbers greater than 4000 cm-1 were modified based on Baranov and Lafferty (2012) and Mondelain et al. (2014) measurements. "

---

## Referee Comment (RC2) · Anonymous Referee #2 · 17 Feb 2018

The manuscript is devoted to measurements of the water vapor self-continuum in the two near-infrared spectral regions. Additional experimental data is presented to those already reported by the same group for other spectral regions. The paper is clearly written, easy to read, and present rather important information about water vapor self-continuum absorption in narrow 3-5 mm radius cells in equilibrium conditions. I think this is a good paper to be published in AMT after accounting for the comments described below.

Main issues:

1. The authors often write about the atmospheric application of their continuum mea-

surements. However, all CRDS measurements by Grenoble group have so far been performed in very narrow (3-5 mm radius) cells where conditions for water vapor can be rather different from those in the atmosphere or in much wider FTS cells. I would not raise this issue for many other gases, but water vapor is a specific case. It is interesting that there is rather close agreement between different measurements of the foreign water vapor continuum, but for the self-continuum, the situation is quite different. Different measurements performed applying different methods give very different results up to more than an order of magnitude.

There is some available evidence, including from satellite measurements (see, for example, the Anonymous Referee #1 comment: "The Richard et al. value was considered in developing recent MT_CKD version, but found to not be consistent with the satellite- and ground-based observations analyzed in Mlawer et al. (2012)"), to suggest that the water vapor self-continuum absorption in the atmosphere can be much stronger than that obtained from 3–5 mm cells in CRDS measurements. One may suggest that water vapor at low pressures may be, for example, depleted of long-living stable water dimers (with a lifetime >= 0.001 s) and/or water nanoclasters in such narrow cells due to their adsorption on cell walls. So, the OFCEAS/CRDS continuum data may be correct for a monomolecular gas, but underestimates atmospheric continuum absorption. I suggest that the authors explicitly recognize this issue in their revision both in their introduction and conclusions.

2. Page 9, lines 14-17: "the FTS values are largely overestimated compared to both MT_CKD and the laser measurements. Compared to OFCEAS, CAVIAR and Tomsk 2015 CS values, reported with a 50 % error bar, are overestimated by a factor of three while the FTS values reported by Baranov and Lafferty with a 22% error bar are overestimated by a factor of two".

This is a somewhat misleading statement. The authors do not take into account the rather strong T-dependence of the self-continuum. They should keep in mind that the OFCEAS data which they show corresponds to about 300 K, while Baranov&Lafferty

data are shown for 311 K, and Tomsk-2015 data, for 287 K. To extrapolate to 300 K, the Tomsk data should be decreased by ∼30%, while Baranov&Lafferty data should be increased by ∼25%.

The T-dependence detected in Baranov & Lafferty's measurements is even stronger than that (see Table 2 in Baranov & Lafferty, JQSRT, 2011 or the left panel of Fig. 7 in the current manuscript). In fact, the Baranov&Lafferty data show a more consistent T-dependence in this window than the present OFCEAS continuum data, and a T-dependence which is as consistent, if not more consistent, with independent FTS data in this same region. So, I personally am not sure which, of OFCEAS or B&L's data, I should trust in this window. The wording here also needs to be more careful, to make clear that is it is specific to near room-temperature.

3. Page 16, lines 5-15: The authors' comments are somewhat misleading in this paragraph, and the authors seem to too easily neglect the results of the FTS measurements. The 0.1–0.2% is a *prediction* of the CRDS measured absorption to the FTS measurement conditions and the argument could be characterized as "circular" – it uses the CRDS measurements to establish that the CRDS measurements must be most reliable! However, the level of continuum absorption measured in FTS experiments in 4 mcr window at close-to-room temperatures was not 0.2%, but 1-1.5 % in Tomsk (2013) and 2–3% in Baranov & Lafferty (2011) measurements. These are rather high values and cannot simply be neglected or characterized just as an "error" (Referee #1 comment). Baseline instability was thoroughly investigated both from pure nitrogen and empty cell absorption before and after the sample measurements, and - at least in Baranov & Lafferty measurements - was several times smaller than the measured continuum absorption:

"Periodic and multiple tests were made to establish the baseline stability. In recording spectra with the cell filled with pure nitrogen up to about 400 kPa (4 atm), no significant systematic changes in baseline caused by mechanical changes in the cell were observed. With much smaller water-vapor pressures, only a small random scatter within

0.3% (one STDV) was observed. A slow drift of the baseline in time was eliminated by averaging spectra of the empty cell recorded before and after the sample spectrum" /Baranov & Lafferty, JQSRT, 112, 1304-13, 2011/

FTS measurements at different pressures were also performed by Baranov & Lafferty (JQSRT, 2011) in contradiction to the implication at lines 11 to 13 of the manuscript.

Again in this paragraph a general statement is made about the quality of FTS measurements without specifically clarifying that the argument is only valid at room temperature as is clear from Figure 7 (and there is evidence, discussed above, that the argument may not even valid at room temperature in atmospheric conditions).

Recent FTS measurements in Tomsk, reported at HRMS-2017, 20–25 August, Helsinki, Finland, were performed at several water vapour pressures at 316 K and a path length of 1000 m, and have detected continuum absorption from 4 to 10% with pressure-squared dependence. They agreed well with Baranov & Lafferty's result for 2500 cm-1 window.

Therefore, I have serious doubts that self-continuum absorption measured in 3-5 mm radius CRDS or OFCEAS cells can be simply applied to atmospheric conditions before the real reasons for such strong disagreement between CRDS/OFCEAS and FTS room-temperature measurements in windows is clarified.

Minor issues:

"Self-continuum" and "Self continuum" – should be unified in the manuscript.

Page 1, line 18: "completing" is exaggerated. First, the 4 micron window is only covered by 3 measurements and at all wavenumbers the range of temperatures measured, compared to what is needed for accurate atmospheric measurements, is sparse. The authors acknowledge this more correctly at p.17 (10).

Page 8, line 6: "on the order of 30 %." According to the Table 1, it is rather about 25%"

Page 8, line 10: "water continuum represents about 25 % of the measured signal (Fig. 2)". According to Fig. 2 it is rather 27-30%.

Figs. 2 & 5: "The contribution of water monomer" is better written as "The simulated contribution of water monomer".

Page 9, Fig.3: Temperatures should be given for OFCEAS measurements and MT_CKD models. This is particularly important given that the OFCEAS measurements vary significantly over a small temperature interval in Fig. 7, and the degree of agreement with MTCKD at 298 K is not found at all at 303 K.

Page 12, line 13: "of the calculated WML contribution calculated as indicated above" – remove the repeated "calculated".

Page 12, line 14: "the WML contribution represents between 18 and 38 % of the total absorption (see Fig. 5)" Again, according to Fig. 5 it is a rather firm 35–38% (nearly independent on pressure). Why do the authors give such uncertain values?

Page 14, Fig 7. The measurements in the 3000 (more properly 3007 cm-1) window do not seem to agree with Table 1. In Table 1, only one point (at 303 K) is over 2.0, and yet 2 points in Fig 7 exceed this value. Might the difference between neighboring points also indicate that the authors are too optimistic in their uncertainty estimates?

Page 17, line18: See also 10.1029/2007GL029259 for evidence of water dimers in the region under consideration in this paper.

---

## Author Comment (AC4) · 1 Mar 2018

-REVIEWER 2

The manuscript is devoted to measurements of the water vapor self-continuum in the two near-infrared spectral regions. Additional experimental data is presented to those already reported by the same group for other spectral regions. The paper is clearly written, easy to read, and present rather important information about water vapor self-continuum absorption in narrow 3-5 mm radius cells in equilibrium conditions. I think this is a good paper to be published in AMT after accounting for the comments described below.

Main issues: 1. The authors often write about the atmospheric application of their continuum measurements. However, all CRDS measurements by Grenoble group have so far been performed in very narrow (3-5 mm radius) cells where conditions for water vapor can be rather different from those in the atmosphere or in much wider FTS cells. I would not raise this issue for many other gases, but water vapor is a specific case. It is interesting that there is rather close agreement between different measurements of the foreign water vapor continuum, but for the self-continuum, the situation is quite different. Different measurements performed applying different methods give very different results up to more than an order of magnitude.

There is some available evidence, including from satellite measurements (see, for example, the Anonymous Referee #1 comment: "The Richard et al. value was considered in developing recent MT_CKD version, but found to not be consistent with the satellite and ground-based observations analyzed in Mlawer et al. (2012)"), to suggest that the water vapor self-continuum absorption in the atmosphere can be much stronger than that obtained from 3–5 mm cells in CRDS measurements. One may suggest that water vapor at low pressures may be, for example, depleted of long-living stable water dimers (with a lifetime >= 0.001 s) and/or water nanoclusters in such narrow cells due to their adsorption on cell walls. So, the OFCEAS/CRDS continuum data may be correct for a monomolecular gas, but underestimates atmospheric continuum absorption. I suggest that the authors explicitly recognize this issue in their revision both in their introduction and conclusions.

If we summarize, the referee hypothesis is that our laser based setups using narrow diameter cells allow for measuring water continua which are of different nature from those relevant for atmospheric applications, because our water sample are missing some contribution due to "stable water dimers and/or water nanoclusters". On the other side, FTS measurements with cells of larger diameters are closer to atmospheric conditions and give access to the "real" atmospheric water continuum. In other words, the MT_CKD continuum which is implemented in many atmospheric codes includes a contribution of "stable water dimers and/or water nanoclusters".....

Without solid scientific arguments we cannot share this opinion.

Several arguments are in contradiction with this hypothesis on the origin of the discrepancies between CAVIAR and Tomsk measurements (some were already given in Ventrillard et al. 2015).

Concerning the influence of the diameter of the cell:

- Our measurements were performed at different periods, by different persons with different setups. Two cavity enhanced techniques were used: CRDS and OFCEAS with cells different by a factor of 2 in diameter (11.5 and 6 mm, respectively). OFCEAS measurements were performed in flow regime while most of the CRDs measurements were performed in static regime. In spite of this variety of experimental conditions we observed a very good consistency of the derived cross section values, for instance in the 2.3 μm window presented below with CRDS and

[Figure]

OFCEAS measurements points highlighted. We believe that this indicates that surface effects do not affect our results.

- There is a very good consistency between the FTS CAVIAR measurements at high temperature with our measurements while FTS used large diameter cells and we use small diameter cells. When one considers the following Fig. , it is hard to imagine that the contents of the water sample are different in the FTS high T cell and in the OFCEAS cell.

[Figure]

- The referee opposes our results to all the previous literature results. As summarized by the following figure below, this is not the case. Our results are relatively close to the results by Burch et Alt obtained with a large cell and also by Bicknell et al with air.
- We agree with the reviewer that the CAVIAR and Tomsk self continuum high value in the window may be due to droplets or aerosol in the FTS cells. The surface/volume ratio of the cell is higher in the case of CRDS and OFCEAS cells but the quality of the vacuum achieved in long cells with volume as high as 10 m$^3$ cannot be at the level achieved with cells with a 10-100 cm$^3$ volume. For instance, on Fig. 2 of Ptashnik et al., 2013, the Tomsk spectra show a "strong contaminating absorption by engine oil at 2800–3000 cm$^{-1}$". The Tomsk FTS spectra shows also a strong band near 2300 cm$^{-1}$ due to $CO_2$ present as an impurity which prevented for continuum retrieval in this region.

[Figure]

According to the referee, the atmospheric water continuum, FTS data (and MT_CKD?) include a contribution of "stable water dimers and/or water nanoclusters" which are destroyed in the CRDS

and OFCEAS cells. If this was the case, we do not see a reason to expect a quadratic pressure dependence of the FTS absorption s ignal. Otherwise it would mean that this additional absorption due to "water nanoclusters" has also quadratic pressure dependence. Is there some scientific justification of such assumption?

Concerning the Richard et al. value at 2491 cm$^{-1}$ which, according Referee 1 was found to not be consistent with the satellite and ground-based observations analyzed in Mlawer et al. (2012), we were not aware about this fact and have no explanation. OFCEAS measurements of the foreign continuum at the same spectral point may help to clarify the situation.

The fact that our values were used to modify the most recent version of the MT_CKD coefficients indicates that for the developers of the MT_CKD model, our continuum values are relevant for atmospheric applications. Let us underline, that if we compare with the 2.5 version of the MT_CKD model which was the version available prior to our first measurements, the overall agreement of our self continuum in the windows is not bad, in any case much better than with CAVIAR and Tomsk values.

Finally, we would like to insist on the fact that the reliability of all our measurements is supported by the careful checking of the quadratic pressure dependence of the absorption signal. No doubt that critical insights on the discrepancy with CAVIAR and Tomsk will be obtained when similar quadratic pressure dependence will be obtained and published for room temperature continuum measurements by FTS in the windows. Up to now, this is not the case. The reported FTS continua rely on single pressure measurements which leaves open hazardous explanations on the origin of the disagreement, including the presence of clusters or droplets which would not be compatible with a quadratic pressure dependence. (Let us mention that convincing quadratic dependence has been demonstrated by the CAVIAR consortium at high temperature (400 K) in the 0.66-2.0 atm range of water pressure (see Fig. 5 of Ptashnik et al. 2011 and Fig. 9 of Shine et al. 2016)). As mentioned in Shine at al. 2016, the situation is particularly confused considering that "For some of the room-temperature FTS measurements, the whole FTS spectrum was adjusted to assumed values of the continuum strength in either higher or lower wavenumber windows."

2. Page 9, lines 14-17: "the FTS values are largely overestimated compared to both MT_CKD and the laser measurements. Compared to OFCEAS, CAVIAR and Tomsk 2015 CS values, reported with a 50 % error bar, are overestimated by a factor of three while the FTS values reported by Baranov and Lafferty with a 22% error bar are overestimated by a factor of two".
This is a somewhat misleading statement. The authors do not take into account the rather strong T-dependence of the self-continuum. They should keep in mind that the OFCEAS data which they show corresponds to about 300 K, while Baranov&Lafferty data are shown for 311 K, and Tomsk-2015 data, for 287 K. To extrapolate to 300 K, the Tomsk data should be decreased by _30%, while Baranov&Lafferty data should be increased by _25%.

We agree with the reviewer that our comparison did not take into account the fact that measurements were performed at different temperatures. Taking into account the T dependence as proposed by the reviewer, Baranov&Lafferty data (311 K) and Tomsk-2015 (287 K) extrapolated at 300 K are about $C_S$= 5.41 and 4.53 ×10$^{-23}$ cm$^2$ molec$^{-1}$ atm$^{-1}$ to be compared to our OFCEAS value at 300 K of 1.96 ×10$^{-23}$ cm$^2$ molec$^{-1}$ atm$^{-1}$. The text has been modified accordingly:

*The CAVIAR $C_S$ value at 296 K and Tomsk 2015 value at 287 K were reported with a 50 % error bar. A rough extrapolation at 300 K leads to $C_S$= 6.3 and 4.53×10$^{-23}$ cm$^2$ molec$^{-1}$ atm$^{-1}$, respectively which should be compared to an OFCEAS value of 1.96 ×10$^{-23}$ cm$^2$ molec$^{-1}$ atm$^{-1}$.The FTS value reported by*

*Baranov and Lafferty at 311 K with a 22% error bar corresponds to about 5.41 ×10$^{-23}$ cm$^2$ molec$^{-1}$ atm$^{-1}$ at 300 K.*

The T-dependence detected in Baranov & Lafferty's measurements is even stronger than that (see Table 2 in Baranov & Lafferty, JQSRT, 2011 or the left panel of Fig. 7 in the current manuscript). In fact, the Baranov&Lafferty data show a more consistent T-dependence in this window than the present OFCEAS continuum data, and a T dependence which is as consistent, if not more consistent, with independent FTS data in this same region. So, I personally am not sure which, of OFCEAS or B&L's data, I should trust in this window. The wording here also needs to be more careful, to make clear that is it is specific to near room-temperature.

We agree that the present OFCEAS measurement of the T dependence at 3007 cm$^{-1}$ is not as convincing as that reported at 2283 cm$^{-1}$ in Campargue et al 2016 and at 2491 cm$^{-1}$ in Richard et al 2017. As illustrated by the Fig below, there is nevertheless a good consistency between the OFCEAS T dependence at the three measurement points. The continuum values and T dependence reported by Baranov and Lafferty are systematically larger than ours. And, as mentioned above, our values are supported by the checking of the quadratic P dependence (lower panels of the Fig.).

[Figure]

3. Page 16, lines 5-15: The authors' comments are somewhat misleading in this paragraph, and the authors seem to too easily neglect the results of the FTS measurements. The 0.1–0.2% is a *prediction* of the CRDS measured absorption to the FTS measurement conditions and the argument could be characterized as "circular" – it uses the CRDS measurements to establish that the CRDS measurements must be most reliable!

We have modified the problematic sentence (For instance, in the center of the 4.0 and 2.1 μm windows, at 15 mbar, the continuum signal to be measured leads to a 0.1-0.2 % light attenuation for

an 500 m absorption pathlength) has been changed to (For instance, in the center of the 4.0 and 2.1 µm windows, at 15 mbar, the continuum measured by OFCEAS and CRDS or predicted by the MTCKD model corresponds to a 0.1-0.2 % light attenuation for an 500 m absorption pathlength).

However, the level of continuum absorption measured in FTS experiments in 4 µm window at close-to-room temperatures was not 0.2%, but 1-1.5 % in Tomsk (2013) and 2–3% in Baranov & Lafferty (2011) measurements.

Yes, of course, this is the reason why there is such difference on the $C_s$ values.

These are rather high values and cannot simply be neglected or characterized just as an "error" (Referee #1 comment). Baseline instability was thoroughly investigated both from pure nitrogen and empty cell absorption before and after the sample measurements, and - at least in Baranov & Lafferty measurements - was several times smaller than the measured continuum absorption: *"Periodic and multiple tests were made to establish the baseline stability. In recording spectra with the cell filled with pure nitrogen up to about 400 kPa (4 atm), no significant systematic changes in baseline caused by mechanical changes in the cell were observed. With much smaller water-vapor pressures, only a small random scatter within 0.3% (one STDV) was observed. A slow drift of the baseline in time was eliminated by averaging spectra of the empty cell recorded before and after the sample spectrum"* /Baranov & Lafferty, JQSRT, 112, 1304-13, 2011/ FTS measurements at different pressures were also performed by Baranov & Lafferty (JQSRT, 2011) in contradiction to the implication at lines 11 to 13 of the manuscript.

For the convenience of the reader we reproduce below the Fig. 4 by Baranov & Lafferty (JQSRT, 2011) to which the reviewer refers:

[Figure]

**Fig. 4.** Measured absorbance in the 1930.9 cm$^{-1}$ micro-window (a, b) and around 2460 cm$^{-1}$ (c, d) at two temperatures. Results observed with an InSb detector are given in circles; dots present measurements with a MCT detector. Solid vertical bars show the water-vapor density limit for a given temperature. Dashed lines give a density cut-off for binary absorption coefficients calculations.

In our opinion, this Figure shows convincing quadratic dependence at 1930 cm$^{-1}$ but not at 2460 cm$^{-1}$. However, 1930 cm$^{-1}$ is not located in the water transparency window. So to the best of our knowledge, there is no experimental report of the pressure squared dependence of the absorption continuum by FTS in the water transparency windows.

Obviously FTS can be performed at different pressures but the possibility to perform rapid pressure ramps at fixed frequency is an advantage of laser-based techniques and we do not see a reason to modify our text:

*The pressure-squared dependence of the absorption signal, systematically fulfilled in the CRDS and OFCEAS studies, provides a crucial test to ensure that baseline variation of the spectra has a gas-phase origin. This is an important advantage of the laser-based approach, to allow monitoring the*

*pressure dependence of the continuum absorption during pressure ramps lasting no more than a few minutes.*

Again in this paragraph a general statement is made about the quality of FTS measurements without specifically clarifying that the argument is only valid at room temperature as is clear from Figure 7 (and there is evidence, discussed above, that the argument may not even valid at room temperature in atmospheric conditions).

We agree that Fig. 7 can be seen as a mutual validation of the high T CAVIAR measurements and our laser-based results. We note that a short path absorption cell was used for the high T CAVIAR recordings and that the measured absorbance were larger as a result of the high water vapor pressure.

Recent FTS measurements in Tomsk, reported at HRMS-2017, 20–25 August, Helsinki, Finland, were performed at several water vapour pressures at 316 K and a path length of 1000 m, and have detected continuum absorption from 4 to 10% with pressure-squared dependence. They agreed well with Baranov & Lafferty's result for 2500 cm-1 window. Therefore, I have serious doubts that self-continuum absorption measured in 3-5 mm radius CRDS or OFCEAS cells can be simply applied to atmospheric conditions before the real reasons for such strong disagreement between CRDS/OFCEAS and FTS room-temperature measurements in windows is clarified.

This last point repeats the Main issue (1) – See our answer above.

We are aware about the results presented by the Tomsk group last August in Helsinki for three pressure values between 35 and 56 mbar at 311 K. We could not find the corresponding publication in recent literature and it is then difficult to discuss these unpublished results. Nevertheless, the authors would like to clarify an important point concerning the treatment of Tomsk FTS spectra. According to Shine at al. 2016, in the treatment of their 2013 and 2015 spectra, the Tomsk group adjusted some of their whole FTS spectra to "assumed values of the continuum strength in either higher or lower wavenumber windows". As indicated in Section 3.2 of Shine at al. 2016, the Tomsk 2013 data rely on the assumption of a negligible continuum value near 9300 cm$^{-1}$ on the basis of Fulghum and Tilleman's measurements while Tomsk 2015 data rely on a different adjustment: *the absolute level of the measurements was adjusted to be the same as MT_CKD2.5 at 287 K in the 2500 cm$^{-1}$ window*. We note that since the V2.5 version the MT_CKD model has been importantly modified in the 2500 cm$^{-1}$ window and then that an adjustment according to the most recent version would lead to important changes on Tomsk2015 data…..

In our opinion, it seems contradictory to claim that the spectra baseline is stable and to perform a global adjustment of the spectra according to assumed value of the continuum. In our experiments, we do not perform such baseline adjustments.

Minor issues:
"Self-continuum" and "Self continuum" – should be unified in the manuscript.

"Self-continuum" is now adopted all along the text.

Page 1, line 18: "completing" is exaggerated. First, the 4 micron window is only covered by 3 measurements and at all wavenumbers the range of temperatures measured, compared to what is needed for accurate atmospheric measurements, is sparse. The authors acknowledge this more correctly at p.17 (10).

Correct. We have changed "completing" to "extending"

Page 8, line 6: "on the order of 30 %." According to the Table 1, it is rather about 25%"

Corrected

Page 8, line 10: "water continuum represents about 25 % of the measured signal (Fig. 2)". According to Fig. 2 it is rather 27-30%.

Corrected (28 %)

Figs. 2 & 5: "The contribution of water monomer" is better written as "The simulated contribution of water monomer".

Done

Page 9, Fig.3: Temperatures should be given for OFCEAS measurements and MT_CKD models. This is particularly important given that the OFCEAS measurements vary significantly over a small temperature interval in Fig. 7, and the degree of agreement with MTCKD at 298 K is not found at all at 303 K.

We have added the temperature information in the caption:

**Figure 3:** Overview comparison of the self-continuum cross-section of water vapor near room temperature.

Solid lines show different versions of the MT_CKD model at 296 K. Experimental results are obtained using different techniques, namely by OFCEAS, (red circles; Campargue et al., 2016; Richard et al., 2017; this work); by FTS from Baranov and Lafferty (2011), light blue circles, from CAVIAR (black squares; Ptashnik et al., 2011a), from Tomsk2013 (orange circles; Ptashnik et al., 2013), from Tomsk2015 (orange circles; Ptashnik et al., 2015) and with a grating spectrograph by Burch and Alt (1984), dark blue circles. Note that the plotted experimental results correspond to different temperature values (see temperature values given in the insert). The temperature of the OFCEAS results are 296.15, and 297.3 and 298.15 K, for Campargue et al., 2016, Richard et al., 2017 and this work, respectively. The 30-50 % error bars on Tomsk2015 values are not plotted for clarity.

Page 12, line 13: "of the calculated WML contribution calculated as indicated above" – remove the repeated "calculated".

Corrected. Thank you.

Page 12, line 14: "the WML contribution represents between 18 and 38 % of the total absorption (see Fig. 5)" Again, according to Fig. 5 it is a rather firm 35–38% (nearly independent on pressure). Why do the authors give such uncertain values?

We report measurements at 4 spectral points between 4495 and 5007 cm-1 (see Table 1). As indicated in the text, the monomer contribution depends on the spectral point. Here is the table with detailed values for the different spectral points.

| WaveNb-- | -- | Total | Mono | Cs | % Mono/Total |
|---|---|---|---|---|---|
| 4995.64 | -- | 1.16671E-23 | 4.14E-24 | 7.53E-24 | 35.5 |
| 4995.63 | -- | 1.17435E-23 | 4.14E-24 | 7.60E-24 | 35.3 |
| 4998.98 | -- | 9.45019E-24 | 1.79E-24 | 7.66E-24 | 18.9 |
| 4998.98 | -- | 9.64154E-24 | 1.79E-24 | 7.85E-24 | 18.6 |
| 5002.05 | -- | 1.06604E-23 | 2.74E-24 | 7.92E-24 | 25.7 |
| 5002.05 | -- | 1.07871E-23 | 2.74E-24 | 8.05E-24 | 25.4 |
| 5006.67 | -- | 1.27251E-23 | 4.80E-24 | 7.93E-24 | 37.7 |
| 5006.67 | -- | 1.27546E-23 | 4.80E-24 | 7.95E-24 | 37.6 |

We have slightly modified the text to be even more explicit.
According to the spectral points, the WML contribution represents between 18 and 38 % of the total absorption *(38 % at 4995.63 cm$^{-1}$, as illustrated in **Fig. 5**).*

Page 14, Fig 7. The measurements in the 3000 (more properly 3007 cm-1) window do not seem to agree with Table 1. In Table 1, only one point (at 303 K) is over 2.0, and yet 2 points in Fig 7 exceed this value.

We thank the reviewer for this good observation. The previous figure was prepared with preliminary Cs values. It has been updated with the values and error bars listed in Table 1:

New version:

[Figure]

Previous version:

[Figure]

Might the difference between neighboring points also indicate that the authors are too optimistic in their uncertainty estimates?

The (smaller) error bars plotted in the previous version did not correspond to those included in Table 1. We hope that the updated plot showing our claimed error bars will convince the reviewer that our uncertainties are not optimistic.

Page 17, line18: See also 10.1029/2007GL029259 for evidence of water dimers in the region under consideration in this paper.

This reference is now quoted:

Paynter, D. J., Ptashnik, I. V., Shine, K. P., Smith, Pure water vapor continuum measurements between 3100 and 4400 cm⁻¹: Evidence for water dimer absorption in near atmospheric conditions, Geophys. Res. Let., 34, L12808, doi:10.1029/2007GL029259, 2007.

*We thank the two reviewers for their detailed analysis and valuable suggestions to improve the paper. Their interest for the reported work and their detailed discussion are particularly appreciated. We hope that the amended version is now suitable for publication in AMT.*

---

## Author Comment (AC3)

*Response to the reviewers of the paper* **"The water vapor self-continuum absorption in the infrared atmospheric windows: New laser measurements near 3.3µm and 2.0 µm" by Loic Lechevallier et al.**
*Atmos. Meas. Tech. Discuss., doi:10.5194/amt-2017-430-RC1, 2018*

REVIEWER 1

The manuscript "The water vapor self-continuum absorption in the infrared atmospheric windows: New laser measurements near 3.3 and 2.0 microns" by Lechevallier et al. presents the latest water vapor self continuum measurements from the Grenoble group, which has been steadily advancing in their quest to perform high-quality self continuum measurements in all near-IR windows of consequence to global energy balance. This new paper analyzes measured values at the high energy end of two windows, regions in which no measurements of equivalent accuracy have been performed previously.

The authors ably present the measurement approach employed and compare their results to previous measurements and the MT_CKD model. The analysis is solid and the paper will make a useful addition to the literature on this subject. Acceptance is recommended, although the authors should implement the improvements suggested below, as well as improve some instances of awkward language (further below).

***Answer:*** The English has been revised all along the text of the new version. Most of the corresponding changes are in orange in the revised version. The changes related to the suggestions of the first and second reviewer are written in blue and green, respectively.

Issues:

1) The uncertainty in the derived self continuum values due to uncertainty in line widths of neighboring strong lines does not seem to be considered. The analysis presented in both regions indicates that the line contribution to the absorption coefficient is significant – for the 3007 cm-1 line it is even larger than the self continuum contribution. Therefore, any substantial uncertainty in the self-broadened line widths would lead to uncertainty in the derived self continuum coefficients. Unfortunately accurate information about the self-broadened line width uncertainty is hard to come by – the best approach might be to assume that it's similar to the uncertainty in the foreign-broadened widths. To develop the aer_v3.6 line parameter file, high-resolution TCCON observations in the near-IR were used to analyze and, if necessary, modify the foreign widths from three sources: HITRAN 2012, Mikhailenko et al., and one from the Regalia group. The differences between the three compilations and widths modified for aer_v3.6 could be as much as 20%. Therefore, for the present manuscript, I recommend the uncertainty provided for the derived self continuum coefficients should be revised to reflect a self line width uncertainty of _20%.

Indeed, in the present work as in our published papers on the water continuum, the reported error bars on the cross sections values do not incorporate the impact of the uncertainty of the line width parameters of the monomer. To the best of our knowledge, this is also the case of all other cross section determinations available in the literature (e.g. from NIST, CAVIAR etc…). Strictly speaking, in the regions where the monomer contribution is significant, like in the present work, the reported continua are dependent of the spectroscopic database used to subtract the monomer contribution. In our understanding, the default database to be used is the HITRAN database and as far as the HITRAN version used is clearly indicated in the text, traceability is insured and there is no reason to propagate the uncertainty on the monomer line parameters to the error bars on the continuum cross sections. The separation of the monomer contribution and of the continuum contribution is a major issue which remains unsolved today. Not only the uncertainty of the line width should be considered but also other factors like the choice of the Voigt profile which is an approximation, the standard ±25 cm-1 cut off and possibly monomer lines missing in spectroscopic databases (for instance, up to the 2012 version, many significant HDO lines were missing in the HITRAN list of water vapor in the near

infrared transparency windows). In that situation, we prefer to associate our continuum values to the HITRAN2016 line parameters of water vapor which insures traceability than to incorporate the self line width uncertainty.

We have nevertheless examined in more details the situation at 3007 cm$^{-1}$ where the self continuum represents no more than about 25 % of the measured continuum signal. A width uncertainty of 20 % on the monomer lines leads to a range of 10-40 % for the continuum value. In fact, most of the monomer contribution near 3007 cm$^{-1}$ is due to the three nearby very strong lines (*).

[Figure]

25 °C and 10 Torr

In HITRAN2016, the line parameters in this region were taken from Loos et al 2017. In their original paper, Loos et al reported error bars less than 1 % for the self broadening coefficient of the 3 lines (see Table , below). This is far below the 20 % error bar suggested by the reviewer and would not impact greatly the error bar on the reported self cross-section values.

| position | intensity | g0s | error on g0s cm-1/atm | % |
|---|---|---|---|---|
| 3004.6863806 | 2.5192e-22 | 0.2493 | 1.5e-03 | 0.602 |
| 3005.4545887 | 8.1126e-23 | 0.1964 | 1.4e-03 | 0.713 |
| 3010.2324498 | 6.4795e-22 | 0.4253 | 2.0e-03 | 0.470 |

We have added the following paragraph at the end of Section 2.2:

Let us underline that, in the present situation where the measured continuum signal is dominated by the water monomer contribution, the retrieved $C_S$ values are strongly dependent on the quality of the monomer line parameters, in particular the self broadening coefficient. Following the standard approach the uncertainty in line parameters of neighboring strong lines was not incorporated in our overall error bar. Our $C_S$ values should then be associated to HITRAN2016 line parameters of water vapor to account for the overall absorption of water vapor in this region. Let us mention that most of the monomer contribution near 3007 cm$^{-1}$ is due to three very strong lines at 3004.686, 3005.454 and 3010.232 cm$^{-1}$ with line intensities in the $8 \times 10^{-23}$-$7 \times 10^{-22}$ cm/molecule range. HITRAN2016 line list of

water vapor reproduces the line parameters obtained by Loos et al. (2017) in the region. For the three strong lines, Loos et al. reported an error bar less than 0.8 % for the self broadening parameters. This is significantly less than the 20-25 % error bar on the reported $C_S$ values.

2. This paper states (pg. 10, line 5 and pg. 15, line 22) that MT_CKD has not yet taken the Richard et al. measurement into account. That leaves the reader with the wrong impression. The Richard et al. value was considered in developing recent MT_CKD version, but found to not be consistent with the satellite- and ground-based observations analyzed in Mlawer et al. (2012). Therefore MT_CKD was reduced as much as possible in that region to reflect the existence of the Richard et al. value but still maintain agreement with these field observations. This paper does a thorough job comparing the new measurements and the previous ones from the Campargue group with prior ones from other teams, but puzzlingly does not present MT_CKD as being measurement-based in this window. This leaves the reader with a misleading impression. The disagreement between the field observations analyzed in Mlawer et al. and the Richard et al. self continuum measurement will hopefully be resolved based on further observations, such as the foreign continuum measurements planned by the Grenoble team (the foreign continuum impacts the analysis of the field measurements) and additional self measurements in this window.

We thank the reviewer for this useful information and have corrected the text: the erroneous statement "the very recent OFCEAS measurement at 2491 cm$^{-1}$(Richard et al., 2017) is not yet taken into account" has been deleted and in the Conclusion, we have repeated the information given by the reviewer:

The Richard et al. value was considered in developing recent MT_CKD version, but found to not be consistent with the satellite- and ground-based observations analyzed in Mlawer et al. (2012). Therefore MT_CKD was reduced as much as possible in that region to reflect the existence of the Richard et al. value but still maintain agreement with these field observations. Further observations and foreign continuum measurements in the center of the 4.0 µm window will hopefully resolve this issue.

Let us note that at this particular spectra point, the t dependence of the MT_CKD continuum differs significantly from the measurements reported by Richard et al. This may also impact have an impact on the level of agreement with field observations:

[Figure]

3. On page 18, last paragraph, the authors have a fair amount of text discussing various analyses of atmospheric absorption based on the FTS measurements. Since this paper and the others from the Grenoble group have basically dismantled the possibility of the self continuum in the near-IR windows being as high as the room-temperature FTS measurements indicated (i.e. significantly larger

than MT_CKD), these high estimates of absorptions can be ruled out. Reading this last paragraph, I expected to have this paper culminate in a clear statement that these high estimates can now be assumed to have been in error. Is there a reason why the authors avoid stating this conclusion?

On page 17, we wrote

"*We believe that the high and mostly constant Tomsk and CAVIAR $C_s$ values of about $4{\times}10^{-23}$ $cm^2 molec^{-1} atm^{-1}$ reported for the 4.0, 2.1 and 1.6 μm windows should be considered as an upper limit due to the detectivity threshold of the FTS approach. In general, the retrieval of a very weak water continuum from a single pressure spectrum is hazardous.* "

We believe that these sentences are sufficiently explicit. As concerned the last paragraph on page 18, we agree that we should have been more conclusive. We have now inserted the sentence suggested by the reviewer:

During very recent years, a set of CRDS and OFCEAS self-continuum cross-sections were obtained over four transparency windows with accuracy as good as a few %. ***These high sensitive laser-based measurements dismantled the possibility of the self continuum in the near-infrared windows being as high as the room-temperature FTS measurements.*** We hope that these results and the forthcoming studies of the foreign-continuum will help to reduce significantly the uncertainty of the impact of the water continuum on the Earth's clear-sky radiation budget.

Typos, minor items:
pg1 line 11 – "in the recent years" should be "in recent years" line 19 – MT_CKD3 should be MT_CKD

Corrected

pg2 line 2 – A fairly small fraction of the solar radiation in the earth's atmosphere is absorbed by water vapor (or absorbed at all). The authors likely mean that 60% of the solar radiation that is absorbed is absorbed by water vapor.

Corrected

line 3 – Since there are all sorts of water vapor absorption bands moving into the visible from the infrared, with weaker vibrational bands spaced with not perfect regularity, the authors might ;want to qualify "located every 1500-2000 cm-1" with a word like ""roughly" or "more-or less".
line 4 – Suggested change for clarity: "These are formed by multitudes of narrow rovibrational absorption lines that are...."

Corrected

lines 18-19 - Suggested change: "This model, which is basically a far-wing line shape model in the window regions, involves a number of ad hoc parameters that have been..." line 20 – "long-infrared" is not typically used.

Corrected

line 22 – The comma after "recently" should probably be removed.

Done

pg3 line 2 – The word "windows" should follow "4.0, 2.1, 1.6 and .25 microns" line 7 –"baseline" is usually one word line 21 – A period is missing at the end of the sentence.

Corrected

pg 4 line 21 – "upstream of" line 31 – Add "with pressure" after "absorption lines".

Corrected

pg 5 line 5 – The reader will have no idea when reading the caption to Figure 1 what is meant by "spectral fit" since its meaning is only clear (the absorption coefficient calculation) when reading about the calculation on page 6 – for example, they might assume it refers to some sort of curve fitting from the data. Perhaps change it to "corresponding spectral fit, as described in the text" or "corresponding absorption coefficient calculation, as described in the text".

Caption modified

pg 9 line 15 – For clarity, add the word "the" between "OFCEAS" and "CAVIAR".

The text has been modified.

pg 10 lines 17-19 – Since 5000 cm-1 is not between 4250 and 4725 cm-1, the wording of this sentence should be changed.

Correct. It is now modified: … sampling of the window by OFCEAS and CRDS from 4250 cm$^{-1}$.

pg 15 line 10 – The use of "law" here implies that the exponential expression is some sort of physical law that the values plotted in Figure 7 should obey. A word like "expression" might be more consistent with the intended meaning.

It is now modified: Nevertheless, near room temperature, the MT_CKD variation is close to the $\exp\left(\dfrac{D_0}{kT}\right)$ function at the three considered spectral points

line 13-14 – "new frequencies of" should be "new frequencies in" (note the missing letter "e")

Corrected

line 15 - V3.2 version is redundant since "V" stands for "version"

Corrected here and everywhere in the text

pg 18 line 5 – It's unclear why there is a reference to Ptashnik et al. here.
According to the MT_CKD web page, the recent modifications to the MT_CKD foreign continuum in the near-IR are due to "Foreign continuum coefficients from 1800-3000 cm-1 were modified to improve agreement with Baranov and Lafferty (2012); in the 1900-2150 cm-1 region, attention was also paid to IASI measurements (Alvarado et al., 2013)....
Foreign continuum coefficients at wavenumbers greater than 4000 cm-1 were modified based on Baranov and Lafferty (2012) and Mondelain et al. (2014) measurements. "

Our statement was not correct. Ptashnik et al., 2012 reported foreign–continuum typically between one and two orders of magnitude stronger than that given previous versions of the MT_CKD model but the recent modification of the MT_CKD coefficients rely on other sources.

The new text is the following:
FTS measurements by Ptashnik et al., 2012 indicated that the foreign-continuum in the near-infrared windows was typically between one and two orders of magnitude stronger than that given in previous versions (2.5 and before) of the MT_CKD model. In the more recent versions, foreign continuum coefficients were increased in the 4.0 and 2.1 µm windows on the basis of measurements by Baranov and Lafferty (2012), IASI (Alvarado et al., 2013) and Mondelain et al. (2014).